# Spatial Spillovers of Financial Risk and Their Dynamic Evolution: Evidence from Listed Financial Institutions in China

**DOI:** 10.3390/e24111549

**Published:** 2022-10-28

**Authors:** Shaowei Chen, Long Guo, Qiang (Patrick) Qiang

**Affiliations:** 1School of Economics, Xi’an University of Finance and Economics, Xi’an 710100, China; 2Great Valley School of Graduate Professional Studies, Pennsylvania State University, Malvern, PA 19355, USA

**Keywords:** spatial spillovers, structural mutation test, spatial econometrics, tail risk network, bonacich centrality, financial impact

## Abstract

This paper investigates the multidimensional spatial effects of risk spillovers among Chinese financial institutions and the dynamic evolution of financial risk contagion in the tail risk correlation network over different time periods. We first measure risk spillovers from financial submarkets to the stock market, identifying five periods using structural breakpoint tests. Then, we construct a spatial error financial network panel model by combining complex network and spatial econometric theory to explore the spatial spillover variability. Finally, we calculate the Bonacich centrality of nodes in the tail risk network and analyze the dynamic evolution of the financial impact path during the different time periods. The results show that the multidimensional spatial spillovers of financial risk among financial institutions are obvious and time varying. The spatial spillovers of financial institutions are positively correlated with the turnover rate and negatively correlated with the exchange rate, interest rate and return volatility. Financial institutions of the same type in the tail risk network display intraindustry risk clustering, and the systemically important institutions identified based on Bonacich centrality differ significantly across time. Moreover, when risk spillovers increase, external shocks’ destructive power and speed of transmission to the network rise.

## 1. Introduction

The rapid development of financial innovation and internet finance has intensified the derealization of the financial industry, and the complexity of financial networks has been exacerbated by mixed operations and business crossovers. Systemically important financial institutions’ defining characteristic has gradually changed from their being “too big to fail” to being “too connected to fail”. At the same time, the deepening of domestic financial integration brought about by the development of financial technology has enhanced spatial links among financial institutions and accelerated the transmission of financial risks.

Scholars have conducted a great deal of research on risk spillovers within financial markets, especially since the global financial crisis in 2008. Scholars are currently focused on risk transmission at three main levels: risk spillovers across international financial markets [1,2,3,4], risk spillovers from international financial markets to domestic financial markets [5,6,7], and risk spillovers across domestic financial submarkets [8,9,10]. Their research methodologies mainly rely on vector autoregression (VAR), conditional value at risk (CoVaR), marginal expected shortfall (MES), and dynamic conditional correlation generalized autoregressive conditional heteroskedasticity (DCC-GARCH) models and their extensions to construct risk spillover indicators among financial markets. Given the effectiveness of the copula function for portraying the tail correlation of financial assets and the superiority of DCC-GARCH models in fitting time series, copula DCC-GARCH models are particularly well suited to portraying the complex dependence structure among financial assets [11,12].

Regarding the causes of financial risk, scholars have provided different explanations focused on three main aspects: information asymmetry, financial innovation, and high leverage. Moral hazard caused by information asymmetry in banks can cause bank runs, forcing banks to sell their assets in a hurry, leading to a decline in asset prices and eventually creating local or global systemic financial risk [13]; this dynamic may have been an important reason for the outbreak of the subprime mortgage crisis [14,15]. Information misalignment can further stimulate speculative behavior, leading to indirect transmission of risk among speculators and eventually spreading to the entire financial system [16,17]. Excessive financial innovation leads to default risk in the loan market, causing liquidity problems among banks, which leads to increased risk in banking-type financial institutions [18,19]. Given its lagging nature, financial regulation is also unable to keep financial innovation within reasonable limits, and the rapid expansion of financial market capacity and soaring investment demand have weakened the robustness of the financial system, thus exacerbating financial risks [20,21]. High leverage resulting from excessive indebtedness in the real sector and credit expansion among financial institutions can increase financial institutions’ contribution to systemic risk [22,23,24]. High leverage among financial institutions around the world can exacerbate banks’ individual business risks, and leverage growth under the pretext of “financial innovation” accelerates the rapid accumulation of bubbles in the stock and real estate sectors, ultimately leading to structural weaknesses in the global financial system, reduced capital liquidity, greater exchange rate volatility, and stagnation in the development of global financial markets [25,26].

All of these explanations point to the key role of financial institutions in the spread of systemic financial risk. Those financial institutions that may cause systemic financial risks are defined by the Financial Stability Board (FSB) as systemically important financial institutions (SIFIs). Wu et al. [27] use a copula CoVaR approach to identify SIFIs in China. The distinctive features of such financial institutions are the large scale of their operations and the high complexity of their business, which can trigger enormous shocks to regional or even global financial systems if a risk event occurs [28,29,30].

The bankruptcy of Lehman Brothers revealed the existence of large financial institutions that were “too big to fail”, but the development of financial innovation has gradually given rise to the phenomena of business and shareholding crossovers among financial institutions, and “too big to fail” has given way to “too related to fail” [31]. The complexity of the relationships between various financial markets and interconnectedness of financial institutions are intertwined, forming structures consistent with those analyzed in complex network theory. Since the small-world and scale-free network models were proposed, scholars have gradually applied complex network theory to study financial risk [32,33,34] and, by combining risk spillover models with network topological features, effectively identify institutions of systemic importance in financial markets [35,36,37].

With the significant increase in spatial links in various global markets, spatial spillovers among different individual units have become the focus of scholars’ attention [38]. Geographic location–based spatial econometric models are uniquely advantageous in identifying spatial spillovers across different financial markets [39,40], but with the in-depth development of spatial econometrics in the field of financial risk, spatial weights based on geographic and economic distance can capture spatial spillovers of financial risk more effectively than location-based spatial weight matrices [41,42,43]. With the deep development of domestic and foreign financial markets, the financial industry and the real economy have become deeply integrated, the degree of financial virtualization has deepened, and new multidimensional spatial spillovers of financial risks across regions, markets, and industries have been observed in the global financial market [44,45]. In contrast, conventional spillover effect methods mainly measure spillover effects based on market information among financial assets. These methods ignore the influence of spatial factors including distance and region on financial risk and do not consider the multidimensional spillover path of financial risk, which lacks the accuracy of identifying spatial spillover of financial risk. It is necessary and important to use new methods to measure the spatial spillover of financial risk, which can capture multidimensional spatial spillover characteristics.

With the duration of COVID-19 epidemic, infectious disease transmission models have received widespread attention from scholars [46,47], which is also applied to the field of financial risk research. The SIR model (where “SIR” stands for “susceptible–infectious–removed”) can effectively portray the evolution of systemic risk in financial networks [48], classifying nodes in the network into three states: healthy (*S*), infected (*I*), and immune (*R*). Healthy nodes that come in contact with infected nodes become infected with a certain probability, while nodes in the infected state transition to the immune state with a certain probability [49]. In the SIR model, the infection rate of healthy nodes is positively correlated with the destructive power of systemic financial risk in the network [50], but the destructive power of infection on the network is not the same for different nodes, and the greater the number of systemically important nodes infected, the more destructive the network is [51,52]. While enhancing the immunity rate of nodes can mitigate contagion of systemic risk [53], it also prolongs the duration of the crisis [54].

In this paper, we first measure the risk spillover from financial submarkets to the stock market using dynamic conditional correlation coefficients and divide the spillovers into five periods using structural breakpoint tests. Then, we combine complex network theory and spatial econometric theory to construct a spatial error financial network panel model based on the tail risk network of financial institutions to explore the variability in the spatial spillovers of financial risk from a multidimensional economic space perspective. Finally, the SIFIs in the network are identified with Bonacich key nodes, and the effect of the dynamic evolution of financial risk on impact paths is analyzed with the SIR model.

The main contributions of this paper are as follows: (1) Existing studies delineating the stages of financial market changes are based mainly on subjective judgments of major events affecting financial markets. We effectively identify abnormal fluctuations in financial markets using objective data based on the structural breakpoints in the dynamic correlations. (2) Existing studies have used spatial econometric models to explore spatial spillovers of financial risks, focusing mainly on spillovers among different economies, markets and regions and mostly limited to a single spatial effect. We take financial institutions as the object of study, construct a measure of economic distance and gravitational spatial weight matrix by considering the geographical distance and financial correlation coefficient, and then analyze the multidimensional spatial spillovers of financial risks across institutions, regions and industries, thereby enriching the existing literature. (3) Using time-varying correlation coefficients, this paper uses structural mutation tests to determine the change stages of financial risk during the sample period. Multidimensional spatial econometric models and complex network theory are combined to describe the time-varying characteristics of financial risk spillover effects, which enriches the research methods of financial risk measures.

The remainder of this paper is organized as follows: Section 2 presents the research methodology, including the measure of risk spillovers from financial submarkets to the stock market, the construction of multidimensional economic space and tail risk networks, and the method of estimating multidimensional spatial econometric regression models. Section 3 presents the results of the empirical analysis, demonstrating the stage changes in the level of risk spillovers from financial submarkets to the stock market, analyzing the variability in the spatial spillovers of financial risk, and discussing the financial risk shock paths using the SIR model. Section 4 concludes this paper.

## 2. Methodology

### 2.1. Measurement of Risk Spillovers in Financial Submarkets

The spillover of financial risks can often be described by the tail correlation of financial time series [55], and when the financial market has drastic fluctuations, the tail correlation will also increase significantly. Engle [56] proposed the DCC-GARCH model. Because it can better describe the volatility spillover effect and information transmission process among financial assets, the DCC-GARCH model has been widely used in describing the dynamic correlation of financial time series. However, the non-normal, spike-back-tailed and tail-dependent features of financial asset return distribution cannot be described by DCC-GARCH model. The copula function is a mathematical method that uses the marginal distribution to determine the joint distribution. It can be used to describe the nonlinear relationship of random variables, and its most important role is to measure the correlation and dependence mechanism between random variables. Meanwhile, previous studies have shown that the t-copula function can effectively describe the interdependence of tail risk between multiple financial time series. The combination of DCC-GARCH model and t-copula can better describe the complex correlation of financial assets. Thereby, we use t-copula-DCC-GARCH model to estimate the dynamic conditional correlation coefficients, which is used to measure risk spillovers in financial submarkets. The detailed derivation process of t-copula-DCC-GARCH model is shown in Appendix A.

With the rapid development of China’s financial market, the impact of major financial events (e.g., financial crisis) on China’s financial markets is becoming increasingly significant. However, most of the current research on the duration of major financial events is based on subjective time points, which largely loses some of the financial market information. Structural mutation refers to the structural change in a time-varying series after a major financial event shock, and this structural change can also lead to changes in interdependence among markets. The Bai & Perron structural mutation test, which classifies structural breakpoints based on structural changes in time series, can be used precisely to identify the occurrence and end of major financial events. Therefore, the identification of structural breakpoints is better than the division based on subjective time points, and can more effectively distinguish the impact of different major events on financial markets. At the same time, Bai & Perron [57] point out that structural breakpoints *m* is generally no more than 5 in most empirical applications; thus, we set the maximum number of mutation points as 4, considering China’s economic environment. Then, the number of mutation points is determined by the minimum Bayesian information criterion. See Appendix B for a detailed description of the Bai & Perron structural mutation test.

### 2.2. Multidimensional Economic Space

#### 2.2.1. Economic Distance Measure

When measuring the distance between financial institutions, the traditional physical distance ignores the multidimension spatial spillover effects. The correlation coefficient based on the stock market index is used as the alternative variable of economic distance, and the combination with physical distance has significant advantages in describing the multidimensional spillover of financial risks [58,59]. Based on this, we introduce the economic distance measure (EDM) describes the spatial correlation among financial institutions. It is usually defined by combining the spatial distance between each two financial institutions and their tail correlations in the financial market. Referring to the research of Li et al. [60], we extend the application of the EDM from the national level to the level of financial institutions. Therefore, the EDM, i.e., *D**_i,j_* between financial institutions *i* and *j*, can be expressed as:(1)$$D_{i,j}=F\left(R_{i,j},d_{i,j}\right)=\sqrt{1-{\left|R_{i,j}\right|}^{d_{i,j}}}\begin{array}{cc} , & D_{i,j}\in \left[0,1\right] \end{array}$$

In Equation (1), *R_i,j_* is the static correlation coefficient measured by the t-copula-GARCH model, representing the tail correlation between financial institutions *i* and *j*, $d_{i,j}\in \left[0,1\right]$; i.e., $d_{i,j}={d^{\prime }}_{i,j}/Max\left({d^{\prime }}_{i,j}\right)$ represents the relative physical distance between financial institutions *i* and *j*. The calculation process of *D_i,j_* is shown in Appendix C.

#### 2.2.2. Gravitational Effect Spatial Weights Matrix

Based on the defined EDM, we construct the gravitational effect spatial weights matrix (denoted as *W*) by introducing the spatial gravity effect of the regional economy and combining the weight index of the geographical region and the weight index of the economic state. The diagonal elements of *W* are all 0, and the off-diagonal elements can be calculated by the following formula:(2)$$w_{i,j}=c_{i,j}\cdot \frac{m_{i}m_{j}}{\exp\left(D_{i,j}\right)}$$
where *c_i,j_* is the control variable. In the process of establishing different types of spatial econometric models, *c_i,j_* can be set as different economic indicators to reflect different economic meanings according to different problems in the financial field. *D_i,j_* represents the EDM between financial institutions *i* and *j,* and *m_i_* represents the proportion of the market value of the *i*th financial institution in the total market value of all financial institutions. Based on the research hypothesis of Arnold et al. [61], the spatial effect among financial institutions is focused on three aspects, and the control variable *c_i,j_* is set equal to 1 to obtain three spatial weight matrices.

(1) The gravitational spatial weights matrix based on the financial market itself, which reflects the general spatial correlation between two financial institutions, is represented by *W_gene_*. *w_i,j_* = 0 for diagonal elements, and *w_i,j_ = c_i,j_m_i_m_j_* for off-diagonal elements.

(2) The gravitational spatial weights matrix based on political administrative relations is denoted as *W_p_*. Set *D_i,j_* = 0 when two financial institutions belong to the same administrative region and *D_i,j_* = 1 when two financial institutions do not belong to the same administrative region. The diagonal elements *w_i,i_* = 0, and the off-diagonal elements $w_{i,j}=c_{i,j}\cdot \frac{m_{i}m_{j}}{\exp\left(D_{i,j}^{P}\right)}$.

(3) The gravitational spatial weights matrix based on the cross-regional spatial correlation of financial institutions’ geographical positions is denoted as *W*_area_. The diagonal element *w_i,i_* = 0, and the off-diagonal element $w_{i,j}=c_{i,j}\cdot \frac{m_{i}m_{j}}{\exp\left(D_{i,j}^{area}\right)}$.

By the newly defined EDM and gravitational spatial weights matrix, we can construct the multidimensional economic space and then capture the transregional and transmarket multidimensional spatial effect. Furthermore, we test the existence of multidimensional spatial spillovers by estimating the spatial econometric regression model.

### 2.3. Multidimensional Economic Spatial Regression Model

The spatial regression model has been continuously developed and comprehensively presented by Cohen-Cole et al. [62]. In the multidimensional economic space, the spatial error model is combined with the stock market to build the financial network panel model with a spatial error term. Assuming that there are *N* financial institutions in the financial network, the spatial error financial network panel model can be expressed as follows:(3)$$y_{i,t}=\sum_{m=1}\beta^{m}x_{i,t}^{m}+\mu_{i}+\xi_{t}+\varepsilon_{i,t}$$
(4)$$\varepsilon_{i,t}=\lambda \frac{1}{w_{i,t}}\sum_{j=1}^{N}w_{ij}\varepsilon_{it}+v_{it}$$
for *i* = 1,2,…, *N*; *t* = 1, 2,…, *T*, where *y_i,t_* is the daily return rate of stock *i* at time *t* in the financial network, *u_i_* is the unit-specific effect of the institution, *ξ_t_* is the unit-specific time effect, $x_{i,t}^{m}$ represents a set of explanatory variables, *ε_i,t_* is the spatial error term, *λ* is the spatial correlation coefficient of the spatial error term, and *w_ij,t_* represents the spatial weight between institutions *i* and *j* in the stock market. For ease of interpretation, Equations (A22) and (A23) are written in matrix form:(5)$$Y_{t}=\beta X_{t}+\mu_{i}+\xi_{t}+\lambda W\varepsilon_{it}+v_{it}$$
where *Y_t_* is an *N* × 1 vector composed of the daily log returns of *N* institutions, *X_t_* is an *N* × *M* matrix composed of *M* variables that may affect the behavior of institutions, and *W* represents the spatial weights matrix, which can be composed of *W_gene_*, *W_area_* and *W_p_*, representing the spatial weight matrix based on the correlations of the market itself, the correlations across regions and the correlations across political administrative regions, respectively. This model is called the spatial error financial network panel model. Its economic meaning shows that the daily return of stocks is affected by the following spatial spillover effect: the general stock weighted return of the financial market (*λ_gene_W_gene_*), the stock weighted return of across regions (*λ_area_W_area_*) and the stock weighted return rate across political administrative regions (*λ_p_W_p_*).

According to the definition of the EDM and the gravitational effect spatial weights matrix *W*, it can be concluded that the EDM is negatively correlated with *w_i_*_,*j*_ and positively correlated with *m_i_m_j_*. From the perspective of *W_gene_*, *m_i_* and *m_j_* (market capitalization of financial institutions) reflect the prior market performance of financial institutions. Since the innate profit-seeking nature, financial institutions always chase financial assets with good market performance and therefore, the returns of financial institutions with good prior market performance usually exhibit the same volatility trend. From the perspective of *W_p_*, the EDM between financial institutions in the same administrative region is smaller than that between financial institutions in different administrative regions. Financial institutions in the same administrative region are susceptible to the same regional policies, which may lead to similar volatility in their daily returns. With regard to *W_area_*, the greater the correlation *R_i_*_,*j*_ between two financial institutions, the closer their business crossover and business transactions are likely to be, and thus, their daily returns will generate the same volatility. Before estimating model (22), we first use the Akaike and Bayesian information criteria (AIC and BIC) to determine whether the unit-specific effects are fixed or random. The test result indicates that the AIC and BIC values of the fixed effects regression are smaller, so the fixed effects panel model is selected. Anselin [63] finds that the fixed effect model with a spatial error term has two problems in estimation: first, there is an endogeneity problem between the spatial error term and the spatial lag term; second, the spatial dependence among different individuals may affect the estimation accuracy of the fixed effect *µ_i_*. Therefore, we use the maximum likelihood (ML) estimation method proposed by Elhorst [64] to estimate Equation (5), the detailed estimation process is shown in Appendix D.

### 2.4. The Tail Risk Network

#### 2.4.1. Rules for the Tail Risk Network

Studies show that the tail correlation among financial institutions rises significantly with crisis events, which enhances the contagion effect of financial risk [65]. Complex networks can effectively describe risk transmission among financial institutions in financial markets [66]. Thereby, we construct the tail risk network with financial institutions as nodes and the tail correlation between financial institutions as edges. The institutional nodes in the tail risk network play the role of both the receiver and the transmitter of risk, which effectively describes the contagious financial risk among financial institutions.

The tail risk association network constructed based on the static correlation coefficient matrix can be represented by the set *g* = (*V, R*), where *V* = {*v*_1_, *v*_2_, *v*_3_,…, *v*_n_} is the set of nodes and element *v*_i_ denotes the *i_th_* financial institution in the network. *R* = {*R_i,j_*} denotes the set of edges with weights in the network (where *R_ii_* = 0).

Since the network with the tail correlation coefficient matrix as the adjacency matrix is a fully connected network and the impact of the less correlated parts on systematic risk can be neglected under external shocks [67], the correlation compression that maintains the network connectivity, i.e., the maximum spanning tree (MST), is used here to effectively demonstrate the overall structure of the network.

The MST can be expressed as *g_T_* = (*V_T_*, *E_T_*), such that *V_T_* = *V*; i.e., the number of vertices of the MST is the same as that in the original graph, which is the same as the definition of the minimum spanning tree, and the defined function *f*(*e*) is:(6)$$f(e)=\left\{ \begin{array}{l} \begin{array}{cc} 1,if & e\in E \end{array}\\ \begin{array}{cc} 0,if & e\notin E \end{array} \end{array}\right.$$

Find *n* − 1 edges in graph *g* to form the MST satisfying *V_T_* = *V*. In calculating the MST, first calculate the graph consisting of 1 minus the weights of all edges, then calculate the minimum spanning tree according to the Prim algorithm (defining the distance between financial institutions as 1 − *R_ij_*, then calculating the minimum spanning tree), and finally calculate 1 minus the weights of the minimum spanning tree; thus, the resulting spanning tree is the MST. The MST reflects the important risk associations among institutions in the tail risk network.

#### 2.4.2. Bonacich Key Node of the Tail Risk Network

In the tail risk association network, the nodes exhibit direct and indirect spatial effects among themselves. Therefore, Bonacich centrality is adopted here to measure the centrality of each node [68]. The Bonacich key node is measured as follows.

The n-dimensional adjacency matrix *G* of network *g* denotes the direct connections in the network, and *G^k^* denotes the indirect connections in the network: $g_{ij}^{k}$ > 0 measures the number of paths of length *K* from *i* to *j* in network *g*, and *G*^0^ = *I*.

Assuming a scalar *λ* ≥ 0 and a network *g*, define the matrix:(7)$$M(g,\lambda )={\left[I-\lambda G\right]}^{-1}=\sum_{k=0}^{+\infty }\lambda^{k}G^{k}$$
where the parameter *λ* is a decay factor; *λ^k^* decreases proportionally with the increase in the path length weights. If *M* (*g*, *λ*) is a nonnegative matrix, its coefficients $m_{ij}(g,a)=\sum_{k=0}^{+\infty }\lambda^{k}g_{ij}^{\left[k\right]}$ calculate the number of paths in network *g* starting at node *i* and ending at node *j*. The weight of a path of length *K* is *λ^k^*. Let 1 denote the n-dimensional vector.

Consider a network *g* with an n-dimensional square adjacency matrix *G* and a scalar *λ* with $M(g,\lambda )={\left[I-\lambda G\right]}^{-1}$ that is well defined and nonnegative. The Bonacich centrality in network *g* with parameter *λ* is $b(g,\lambda )={\left[I-\lambda G\right]}^{-1}\cdot 1$.

The Bonacich centrality of node *i* is $b_{i}(g,\lambda )=\sum_{j=1}^{n}m_{ij}(g,\lambda )$, which calculates the total length of all paths starting from node *i* in network *g*, i.e., the sum of all loops *m_ii_*(*g*, *λ*) from node *i* to node *i* and the total number of paths from node *i* to node *j*(*j* ≠ *i*), denoted as:(8)$$b_{i}(g,\lambda )=m_{ii}(g,\lambda )+\sum_{j\neq i}m_{ij}(g,\lambda )$$

Additionally, referring to the spatial effect multiplier (*φ*) used by Cohen et al. [62] to portray the sensitivity of the tail risk network to shocks and the magnitude of shocks, *φ* = 1/(1 − *λ*) indicates that shocks spread rapidly in the network with the parameter *φ* as a multiplier. The larger *φ* is, the more sensitive the system is to shocks, and the greater is the intensity of shocks.

In summary, the parameter *λ* is the spatial correlation coefficient among the financial institutions, capturing the spatial effects and the strength of the interactions between financial institutions. The magnitude of *λ* reflects the extent to which the network is subject to shocks. Therefore, the Bonacich key point measure in a multidimensional economic space exactly captures both direct and indirect spatial effects between institutions.

## 3. Empirical Study and Results

### 3.1. Data Description

Since China’s public listing financial market is still in its developing stage, most financial institutions were listed only within a few years. According to the information disclosed by the China Securities Regulatory Commission, as of the third quarter of 2021, there were 126 public listed financial institutions in China, including 41 banks, 34 securities institutions, 7 insurance companies, and 44 diversified financial institutions. Considering the requirement of the size and the timeframe of the data set, we select 56 financial institutions listed before January 2010 as the sample, including 14 banks, 11 securities companies, 5 insurance companies, and 26 diversified financial institutions, which is a highly representative sample., as shown in Appendix E
Table A1. The time window is from 4 January 2010, to 18 April 2022, and the whole sample period covers as many market conditions as possible, such as bull market, bear market, and market recovery periods, in line with the study objectives. Considering that listed companies may suspend their trading, for the closing price of a company during suspension periods, we use the closing price on the last day before the suspension. Table 1 shows the descriptive statistics of the log returns of listed financial institutions. Here, the continuous log returns of each listed financial company are calculated using the formula *r_i,t_* = log(*P_i,t_*) − log(*P_i,t−1_*), where *P_i,t_* denotes the closing price of institution *i* at time *t*, *P_i,t_*_−1_ denotes the closing price of institution *i* at time *t* − 1, and all data are obtained from the Wind database.

According to the descriptive statistics in Table 1, the mean daily return of the 56 financial institutions is negative, while the rest of the statistical indicators are positive; each financial institution showed weak business conditions after 2010. In terms of the standard deviation, banking financial institutions have the lowest volatility, followed by insurance and securities institutions, and diversified financial institutions have the highest volatility. This is mainly because most banking financial institutions adopt a more prudent investment strategy, while securities and insurance companies mainly adopt a more aggressive investment approach and other types of financial institutions are more focused on short-term interests. The investment behavior of the latter is more rapid and decisive, with a higher sense of the market, and therefore, their returns are more volatile. Meanwhile, except for a limited number of financial institutions, all return series exhibit skewed negative distributions, have kurtosis means over 3, and have maximum values over 300. This indicates that financial institutions are extraordinarily sensitive to external shocks such as financial risks; each return series exhibits an abnormal, spiky, thick-tailed distribution. Finally, the results of Jarque-Bera (JB) statistic and ADF test show that all the return series obey a Gaussian distribution and are stationary at the 1% significance level.

### 3.2. Dynamic Correlation Analysis and Breakpoint Detection

The correlation between financial sectors increases significantly when there is violent turbulence in the financial market [69]. Therefore, using the Shanghai-Shenzhen 300 financial index (denoted as the *Fin300*) to represent the development of China’s financial sector market and the Shanghai-Shenzhen 300 index (denoted as *HS300*) to represent the development of China’s stock market, we can use the correlation between the two markets to measure the risk spillover from China’s financial sector market to the overall economy (The results of Jarque-Bera (JB) statistic and ADF test show that *Fin300* and *HS300* obey a Gaussian distribution and are stationary at the 1% significance level). Since stock returns have nonnormal distribution characteristics such as spikes, thick tails, autocorrelation and asymmetry, GARCH models can effectively portray the return on financial assets and the simplest GARCH (1,1) model captures the characteristics of the return on financial assets [70,71]. Before estimating the return on each asset, it is necessary to estimate an autoregressive moving average model (ARMA) of asset returns, which captures the asymmetry in volatility. According to the AIC criterion, the optimal lag order of the ARMA model is chosen as (2,2). A vector autoregressive moving average GARCH (ARMA-GARCH) model is used to fit the two markets with an ARMA (2,2)-GARCH (1,1) model, and the residuals are set to be t-distributed due to the thick-tailed property of the data. In addition, since the t-Copula function is free from the assumption of normal distribution, it can separate the marginal distribution from the joint distribution and capture the nonlinear, asymmetric tail correlation between variables. After obtaining the residual term, we obtain the dynamic conditional correlation coefficients of *Fin300* and *HS300* by fitting a t-copula-DCC-GARCH model.

Figure 1 shows the trend of the dynamic conditional correlation coefficients from 5 January 2010, to 18 April 2022, showing a clear time-varying characteristic. In general, the dynamic conditional correlation coefficients of the two markets generally fluctuate between 0.6 and 1 and are susceptible to impacts from extreme events, with larger fluctuations occurring in 2010–2013, 2015–2017, and 2018–2019. As a result of the European debt crisis at the end of 2009, China’s foreign exchange market experienced dramatic fluctuations in 2010, with the RMB appreciating significantly relative to the currencies of European countries. Moreover, the EU is an important economic and trade partner of China, and thus, the instability in the eurozone exacerbated the volatility of China’s financial markets. In June 2013, China’s financial markets experienced a liquidity crisis called the “money shortage”, followed by a period of record high interbank repo rates and interbank lending rates, which subsequently led to consistently lower stock markets and significantly increased risk spillovers in the financial market. Before June 2015, stock market indices climbed one after another, but then came the crash of thousands of stocks in the Chinese A-share market, with the Shanghai Composite Index falling more than 32.11% in the 17 trading days after June 12. From 2017 to 2018, the government strengthened the regulation of the financial sector, and the Financial Stability and Development Committee under the State Council (FSDCSC) was established in 2017. At the same time, the introduction of new regulations on capital management not only broke the rigidity of payment structures but also reduced the leverage ratio of each sector. During this period, liquidity in China’s financial markets tightened, and the correlation between the financial sector and equity markets decreased significantly. However, the ensuing debt defaults and a wave of person-to-person (P2P) platform failures led to a significant increase in financial risk spillovers. With the breakdown of trade talks between China and the US in 2019, China’s foreign trade suffered a major setback. Influenced by internal and external factors, the overall performance of the RMB/USD exchange rate first rose, then fell, and finally rose again, with high volatility, leading to a rise in risk spillovers in financial markets.

With the conclusion of the first-stage economic and trade agreement between China and the United States on 13 December 2019, risk spillovers in the financial market decreased to a certain extent. However, the ensuing nationwide COVID-19 pandemic, which hit both the real economy and financial markets, led to high risk spillovers in financial markets again. After entering the post-epidemic period, risk spillovers in the financial market decreased significantly. Although they rose in some cases, they were generally smaller than in the previous period. However, a significant rise could also be observed after 2022, caused mainly by the volatility in international energy markets triggered by the Russia-Ukraine conflict.

The method proposed by Bai & Perron [72] for global minimization of the residual sum of squares is used to test the mean mutation of the dynamic conditional correlation coefficients. In viewing of the timing of the extreme events mentioned above, the number of mutation points is set to 4 here. The changes in risk spillovers in China’s financial markets during the sample period can be divided into five stages, as shown in Table 2.

### 3.3. Multidimensional Spatial Effect Test

Before we estimate the multidimensional spatial regression, it is necessary to test whether there is a spatial effect on the stock returns of listed financial companies and to check the validity and adaptability of the newly constructed gravitational spatial weight matrix.

In spatial econometrics, the main measures of spatial correlation include the global Moran’s I index, *global G* coefficient, Geary C coefficient, LMlag, LMerro and join-count statistic. The classical global Moran’s I index is applicable only to cross-sectional spatial econometric models and is no longer valid for spatial econometric panel models. Here, the spatial weight matrix in the classical global Moran’s I index model is blocked: $K=I_{T}⊗W$, where ***K*** is the *NT × NT* blocked diagonal matrix, ***I****_T_* is the *T*-order identity matrix, ***W*** is the *N*-order spatial weight matrix, and ⊗ denotes the Kronecker product. The improved global Moran’s I can be extended to test for spatial effects in panel data, as calculated by the following equation:(9)$$\begin{array}{cc} Moran’s & I \end{array}=\frac{N\sum_{i=1}^{N}\sum_{j=1}^{N}k_{i,j}\left(x_{i}-\bar{x}\right)\left(x_{j}-\bar{x}\right)}{\sum_{i=1}^{N}\sum_{j=1}^{N}k_{i,j}{\left(x_{i}-\bar{x}\right)}^{2}},\begin{array}{cc} Moran’s & I \end{array}\in [-1,1]$$
where *x_i_* is the attribute value of institution *i* and *k_i_*,*_j_* is the correlation degree between spatial units *i* and *j*, i.e., the blocked spatial weight matrix. The closer the global Moran’s I index value is to −1, the stronger is the negative correlation between regions; the more the value tends to 1, the stronger is the positive correlation; if the value is equal to 0, there is no correlation.

Geary C is another indicator commonly used in spatial autocorrelation tests that emphasizes the dispersion between observations. The value of Geary C ranges from 0 to 2. If the value is greater than 1, it indicates a negative correlation; if the value is equal to 1, it indicates no correlation; if the value is less than 1, it indicates a positive correlation. In addition, Anselin et al. [73] propose the Lagrange multiplier (LM) test of spatial correlation (LMlag and LMerro test) and compare the significance level of the LM spatial lag term and the LM spatial error term. That is, a nonspatial panel model is used for ordinary least squares (OLS) regression analysis, and then the estimated results are subjected to the LM test. The original hypothesis of the LM test is that there is no spatial correlation among the residuals. Here, we select the global Moran’s I index, Geary C index, LMlag, LMerro, robust LMlag and robust LMerro to test listed financial institutions’ spatial effect on stock returns (Because the time dimension of the daily return rate of financial institutions in the sample period is too large, the software cannot effectively calculate the Kronecker product. In this paper, the monthly return series of financial institutions from 2005 to 2020 are used to calculate the block diagonal matrix *K*(*NT* × *NT*), where *N* = 56 and *T* = 148). The test results are presented here based on the full sample data, as shown in Table 3.

From the test results in Table 3, we find that (1) there are significant spatial effects in all three types of spatial weight matrices. The Moran’s I index values are all greater than 0, and all pass the *Z*(*I*) significance test. All Geary C test results are less than 1. The Moran’s I and Geary C test results based on *W_gene_*, *W_P_* and *W_area_* are all statistically significant at the 1% level. This indicates that there is a significant multidimensional effect and a positive regional correlation for the stock returns of listed financial institutions. (2) All three types of spatial weight matrices have significant spatial lag effects and spatial error effects, and the spatial error model (SEM) is statistically significantly better than the spatial lag model (SAR), so the spatial error model is chosen. From the test results, the largest spatial correlation appears for the relationship between regional administrative organizations. The difference between the spatial correlation based on the general market and across regions is not significant. This can also be explained by the frequent fluctuations in China’s financial market since 2010, while the central and local governments have made important contributions to financial stability.

### 3.4. Spatial Spillover Effect Analysis with the Multidimensional Economic Spatial Regression Model

The identification of Bonacich key nodes in the tail risk network requires the calculation of spatial spillovers of financial risks; i.e., it is necessary to analyze what factors influence financial institutions’ spatial risk spillovers. The maximum likelihood method proposed by Elhorst [64] is used to estimate Equation (A22), and the spatial error financial network panel model is constructed in five subperiods based on the three spatial weight matrices described previously. Referring to Zhang et al. [74] and Weng et al. [75], here, we use the stock return of financial institutions as the dependent variable, the turnover rate as the core explanatory variable, and the exchange rate and interest rate as control variables. Meanwhile, we introduce two dummy variables (low and high volatility) to analyze the sensitivity of returns to financial market volatility.

The daily returns of all financial institutions are fitted by an ARMA(p,q)-GARCH model. For each trading day, the returns of the financial institutions are divided into four groups according to their estimated standard deviation values from lowest to highest. The first group within each trading day is called the “low-volatility group”, the fourth group is called the “high-volatility group”, and the middle two groups are used as reference groups. The high-volatility dummy variable is set to 1 for the high-volatility group and 0 for the low-volatility group; the low-volatility dummy variable is set to the opposite values; and the values of the two middle reference groups remain unchanged. Table 4 defines all the explanatory variables.

Table 5 gives the estimation results under different spatial weight matrices during the five subperiods. Considering it along with Figure 1, we find that the time-varying correlation (dynamic conditional correlation coefficients) between *Fin300* and *HS300* has a trend consistent with that of the spatial spillover effect (*λ*) of risk over the subperiods. With an increase (or decrease) in the time-varying correlation, the spatial spillovers of financial risks also increase (or decrease). Observing the trend of the time-varying correlation coefficient, we find that the coefficient for stage 4 is the largest and the corresponding spatial spillover effect the strongest, while the coefficient for stage 3 is the smallest and the corresponding spatial spillover effect the weakest. Specifically, the time-varying correlation rises rapidly when extreme events occur and the spatial spillover effect of financial risk also increases more significantly. Observing the changes in the fourth and fifth periods, we can see that the time-varying correlation and the spatial spillover effect increase simultaneously during the outbreak of the COVID-19 epidemic while they decrease simultaneously during the post-epidemic period. In addition, the time-varying correlation and spatial spillover effects change significantly before and after the outbreak of the “European debt crisis”.

From the perspective of the spatial weight matrix, the estimation results show that the spatial spillovers of risks based on *W_p_* are the strongest (except for the first period) and that the differences in the spatial spillovers of risks based on *W_gene_* and *W_area_* are not significant. The Chinese government plays an important role in promoting economic development. In addition, the economic policies of local governments have a strong influence on economic development within the same region. Hence, the spatial spillover effect based on *W_p_* is the strongest. Meanwhile, due to the fact that *W_gene_* and *W_area_* portray the spatial spillover effect of financial risk in terms of general market correlation and economic distance, which mainly reflects the spatial spillover effect in terms of market factors, their spatial spillover effects are not significantly different.

In terms of the spatial spillover effects in each period, the trend first shows a decrease, then an increase, and finally a decrease again. Specifically, the spatial spillovers of financial risks are strongest in stage 4 (which includes the US–China trade conflict and the COVID-19 epidemic shock), followed by stage 1 (which covers the financial crisis and the European debt crisis), followed by stage 2 (which includes the money shortage and stock market crash in China) and stage 5 (which includes the post-epidemic period). Finally, stage 3 (the calm period) has the smallest risk spillovers. According to the time-varying characteristics of spatial spillover effects, the differences in spatial spillover effects at different stages are mainly influenced by government behavior and extreme events. The COVID-19 epidemic has had a dramatic impact on financial markets, which led to a large set of government interventions, and hence, the spatial spillover effect is strongest in the fourth period. The “European debt crisis”, as a global event, had a significant impact on China on its import/export business. Therefore, the spatial spillover effect in the third period is also stronger. During the period of “money shortage” in 2013 and the “stock market crash” in 2015, the government restricted the excessive boom of the financial market through policy measures, resulting in a rapid fall of the Chinese stock market within a short period of time. However, the impact was mainly on small and medium-sized financial institutions and the impact on large state-owned financial institutions was limited, which makes the spatial spillover effect of risk relatively small. In the post-epidemic era, although there were no extreme events in the domestic and international financial markets, the resurgence of the epidemic also had a large impact on the market. Hence, the spatial spillover effect did not decline significantly. Finally, in the third period, the government acted in line with market expectations and extreme events did not occur, which caused the spatial spillover effect declined significantly.

Regarding the estimated regression coefficients of the main explanatory variables, the turnover rate (*turnover*) is significantly and positively correlated with the log return rate (*r*), indicating that the higher the demand for stocks is, the higher the return rate. The dummy variable *lvar* is not significantly correlated with the log return rate (*r*), whereas *hvar* shows a significant negative correlation, indicating that in China’s financial market, stockholders and stock traders are not sensitive to stocks with low volatility but are more sensitive to stocks with high volatility. Therefore, when stock volatility is high, stockholders and stock traders have less desire to buy them because of the higher risk, and thus, their returns are smaller. This also indicates that most shareholders and stock traders in the stock market are rational investors and will actively avoid high-risk investments [76].

In terms of the estimated coefficients of the control variables, *Erate* has a significant negative correlation with the log return rate (*r*) starting from stage 4, due to the trade conflict between the US and China, which exacerbated the volatility of the RMB against the USD. The cross-country capital flows accompanying the US sanctions against Chinese firms further exacerbated the volatility of the Chinese stock market. The change rate of the interest rate (*DR001*) exhibits a significant negative correlation with the log return rate (*r*) for stages 1 to 4, while it does not correlate significantly with the stock returns of financial institutions in stage 5. This suggests that the prolonged extension of the COVID-19 epidemic has led to a significant amplification of the original poorly regulated Chinese interest rate mechanism and shows that financial institutions are more sensitive to economic uncertainty than in previous stages.

In addition, a positive correlation between the spatial effect multiplier *φ* and the spatial spillover effect *λ* is observed in the estimation results. The spatial effect multiplier *φ* increases rapidly in the presence of extreme external event shocks, including in stages 1, 2 and 4, while it shows a rapid decrease for the stable period of stage 3.

### 3.5. Tail Risk Network and BONACICH CEntrality

We construct a tail risk correlation network based on complex network theory, with the financial institutions as network nodes and the tail correlation of each financial institution’s return as the connected edge. To identify the SIFIs in view of the spatial spillovers of financial risk, we introduce the spatial spillover effect *λ* to calculate the Bonacich centrality of the network nodes. The maximum spanning tree–based tail risk network is given in Figure 2, where Figure 2a–e show the MST network from stages 1 to 5 in order and the size of the nodes reflects the Bonacich centrality score. In addition, the estimation results based on different spatial weight matrices show that the spatial econometric model based on the spatial weight matrix *W_area_* fits best. Therefore, the spatial spillover effect (*λ*) based on *W_area_* is incorporated into the Bonacich key node measurement and the dynamic evolution process of the SIR model.

In general, the risk spillover is larger for banking-type financial institutions in the first period, and from the second period onward, the risk spillover is larger for diversified financial institutions. The risk spillover effect of securities-based financial institutions and insurance-based financial institutions is smaller. From the maximum spanning tree network of the five subperiods, the risk spillovers among financial institutions of the same type are more obvious, which indicates obvious intraindustry risk aggregation characteristics among the four types of financial institutions (banking, securities, insurance, and diversified financial institutions) in China.

The network centrality feature is a key characteristic of tail risk networks, and Bonacich centrality introduces spatial spillovers of financial risk into the centrality measure, the magnitude of which indicates the degree of risk spillover caused by a node in the network, reflecting the degree of importance of the node in the tail risk correlation network. The top 10 listed financial institutions in terms of Bonacich centrality for the five subperiods are given in Table 6.

According to the measurement results in Table 6, Bonacich centrality can effectively identify systemically important institutional nodes under different financial risk shocks. When international financial risk shocks (e.g., the European debt crisis) occur, large state-owned financial institutions (e.g., Industrial and Commercial Bank of China, Bank of Communications) can effectively withstand external shocks, so their Bonacich centrality in the network is higher. When the domestic financial market fluctuates drastically (as during the money shortage and stock market crash in China), because small and medium-sized financial institutions are more engaged in risky investments. their scale is smaller, and their business types undiversified, their ability to resist risks is weaker, and the degree of risk spillover is stronger. Therefore, their Bonacich centrality is higher. In addition, Bonacich centrality can identify financial institutions experiencing business crises. In the third, fourth and fifth periods, “ST.DLT” and “ST.DPF” with the letters ST are financial institutions facing delisting, with relatively poor operating conditions and high Bonacich centrality.

In addition, in terms of temporal evolution, the nodes of SIFIs in China’s financial market change over time. With the diversification of China’s financial market, commercial banks, insurance institutions and securities institutions play a dominant role in the financial market. However, based on the contagion perspective of financial risk, the risk spillovers from diversified financial institutions are significantly higher than those from traditional financial institutions in the face of shocks from external events. When there are changes in the domestic financial market environment, such as the 2015 stock market crash in the second period, the China-US trade conflict and the COVID-19 epidemic in the fourth period, the spatial spillovers of financial risks from diversified financial institutions (such as DXC and DAX) are stronger than those from traditional financial institutions.

### 3.6. The Dynamic Evolution of the SIR Model

The SIR model is derived from the infectious disease model in the medical field and is now widely used in research related to risk contagion in financial markets. We classify listed financial institutions into susceptible financial institutions (*S*), infected financial institutions (*I*), and permanently immune financial institutions (*R*). *S* indicates that the specified type of financial institution is vulnerable to financial shocks but is not currently exposed to them. *I* indicates that the specified type of financial institution has been exposed to financial shocks and has the ability to transmit risk, and *R* indicates that the specified type of financial institution will not be exposed to financial shocks or has withdrawn from the financial network. Moreover, assume that the probability of transforming type *S* financial institutions into type *I* financial institutions is *α* and that the probability of transforming into type *R* financial institutions is *β*.

*S*(*t*), *I*(*t*) and *R*(*t*) denote the number of financial institutions in the three scenarios *S*, *I* and *R*, respectively. According to the study sample, *S*(*t*) + *I*(*t*) + *R*(*t*) = 56. We analyze the dynamic evolution of the impact path under different spatial effect parameters based on the five subperiods. In this process, the transition probability *α* is equal to the spatial correlation coefficient *λ_area_*, and the transition coefficient *β* = 0.01. Assume that only one financial institution node in the financial network is hit at the beginning of the period, i.e., *I*(0) = 1, *S*(0) = 56. The dynamic evolution of the impact path of systemic financial risk is shown in Figure 3, and the spatial correlation coefficients from Figure 3–e are 0.809, 0.704, 0.405, 0.833, and 0.691, respectively.

Figure 3 shows that there is significant variability in the impact of different spatial correlation coefficients on the network. The larger the spatial correlation coefficient is, the stronger the impact on the network, which is shown in Figure 3 as the larger slope of the *I*(*t*) curve. When the spatial correlation coefficient in the third period is 0.405, *t* = 25 is the time of collapse of the network, and when the spatial correlation coefficient in the second period is 0.833, *t* = 9 is the time of collapse of the network. In addition, we find that the maximum number of infected nodes of the network varies under different spatial effects. The maximum number of infected nodes is 53 in the first period when the spatial correlation coefficient is 0.809 and 50 in the third period when the spatial correlation coefficient is 0.405. This further indicates that as the spatial spillover effect of financial risk increases, its spillover speed and destructive power also increase significantly.

Bonacich key nodes have time-varying characteristics, and we need to strengthen risk management and risk control for the industries where key nodes are clustered in different periods, which can reduce the spatial spillover effects of financial risks. Specifically, when international financial crises occur, China’s large state-owned commercial banks and others bear most of the risk. In the event of severe turbulence in the domestic financial market, small and medium-sized diversified financial institutions become the main risk spillover institutions. Therefore, by identifying the types of financial risks and then targeting the main risk spillover industries, the spatial spillover of risks can be reduced. This can reduce the probability of risk contagion in the SIR model and decrease the slope of the *I*(*t*) curve.

## 4. Conclusions

This paper discusses spatial spillovers of financial risk and their dynamic evolution from the perspective of financial institutions. Using daily return data from the Chinese financial market and 56 financial institutions from 4 January 2010, to 18 April 2020, we first measure the dynamic conditional correlation coefficients of the China 300 Financial Index and the CSI 300 Index based on the t-copula-DCC-GARCH model to measure the risk spillovers from the financial sector to the stock market. To analyze the time-varying characteristics of the financial risk spillovers, we divide the dynamic conditional correlation coefficients into five periods based on the Bai & Perron structural breakpoint test. Then, combining complex network theory and spatial econometric theory, we construct a spatial error financial network panel model based on the tail risk network of financial institutions to further explore the variability in the multidimensional spatial spillovers of financial risks. Finally, SIFIs in the network are identified using Bonacich key nodes, and the dynamic evolution of financial risk shock paths is analyzed. The findings of the study are as follows.

First, the correlation between the financial sector submarket and the stock market has obvious time-varying characteristics, with dynamic conditional correlation coefficients ranging between 0.6 and 1, indicating that the financial sector submarket has strong risk spillovers with the stock market. Based on the Bai & Perron structural breakpoint test, financial risk spillovers and their changes can be divided into five periods, revealing obvious phase characteristics and a vulnerability to extreme events.

Second, there are significant multidimensional spatial spillovers of financial risk among financial institutions. The test results of the spatial spillover effects show that the strongest spatial spillovers are based on *W_p_* while the spatial spillovers based on *W_gene_* and *W_area_* are not significantly different. The estimation results of the established spatial error financial network panel model show that the model based on *W_area_* has the best estimation results. Specifically, regarding the explanatory variables, financial institutions’ returns are significantly and positively correlated with the turnover rate, the high-volatility dummy variable has a significant negative effect on returns, and the low-volatility dummy variable has a nonsignificant effect on returns in all subperiods. In addition, there is a significant change across phases in the effect of the interest rate and exchange rate on returns. Prior to the fourth period, returns are significantly negatively correlated with interest rates, but after it, they are not correlated, while after the fourth period but not before it, returns are significantly negatively correlated with the exchange rate.

Third, an MST network is constructed based on the tail risk network, and the study finds that there is significant intraindustry risk clustering in Chinese financial institutions such as banking, insurance, securities, and diversified financial institutions. Bringing the *W_area_*-based spatial spillover effects into the Bonacich centrality measure, we find that there are significant differences in the systemically important nodes of the network within the five subperiods. In the first period, large state-owned commercial banks mainly act as the key nodes of the network. Starting from the second period, the systemic importance of large state-owned commercial banks decreases, while that of diversified financial institutions rises continuously. Moreover, as the spatial spillovers of financial risks increase, the destructive power and speed of shocks caused by financial risks also rise.

In summary, policymakers should focus on the following three areas when conducting financial risk prevention. First, they should differentiate between international and domestic financial risks. In the face of international financial risks, policymakers need to focus on the risk level of large financial institutions, and when domestic financial markets are volatile, small and medium-sized diversified financial institutions need to be guided to strengthen their risk management. Second, the multidimensional spatial spillovers of financial risks must be considered. Based on the cross-market, cross-industry and cross-regional contagion characteristics of financial risks, comprehensive risk prevention measures should be formulated, focusing on preventing cross-regional risk spillovers. Finally, the interest rate and exchange rate transmission mechanism should be unblocked to ensure that monetary and fiscal policies can be effectively transmitted to the financial market.

## Figures and Tables

**Figure 1 entropy-24-01549-f001:**
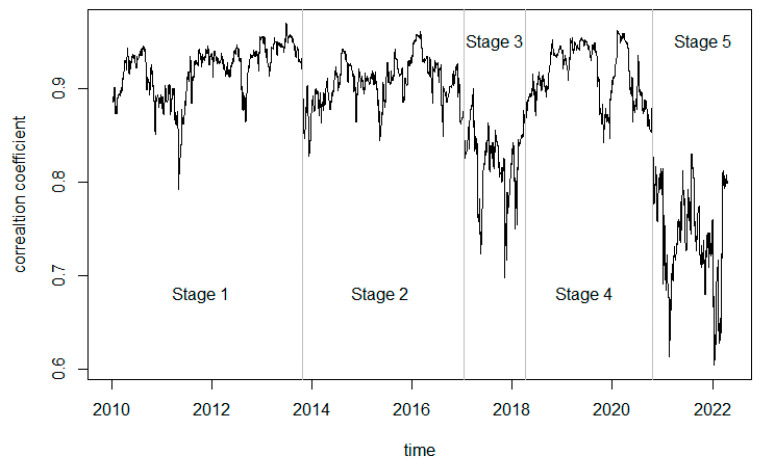
Trend of the dynamic conditional correlation coefficient.

**Figure 2 entropy-24-01549-f002:**
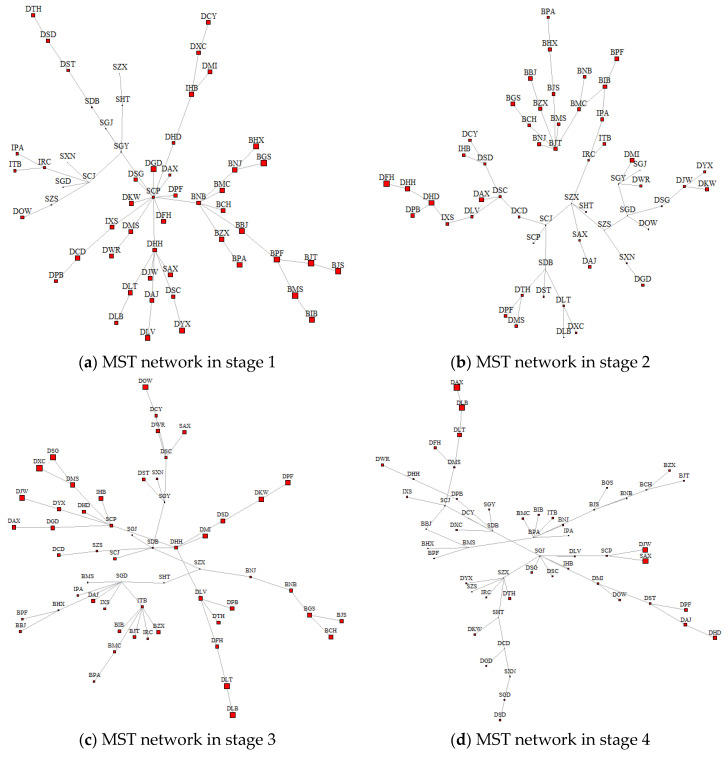
Dynamic evolution of the MST network during the five subperiods.

**Figure 3 entropy-24-01549-f003:**
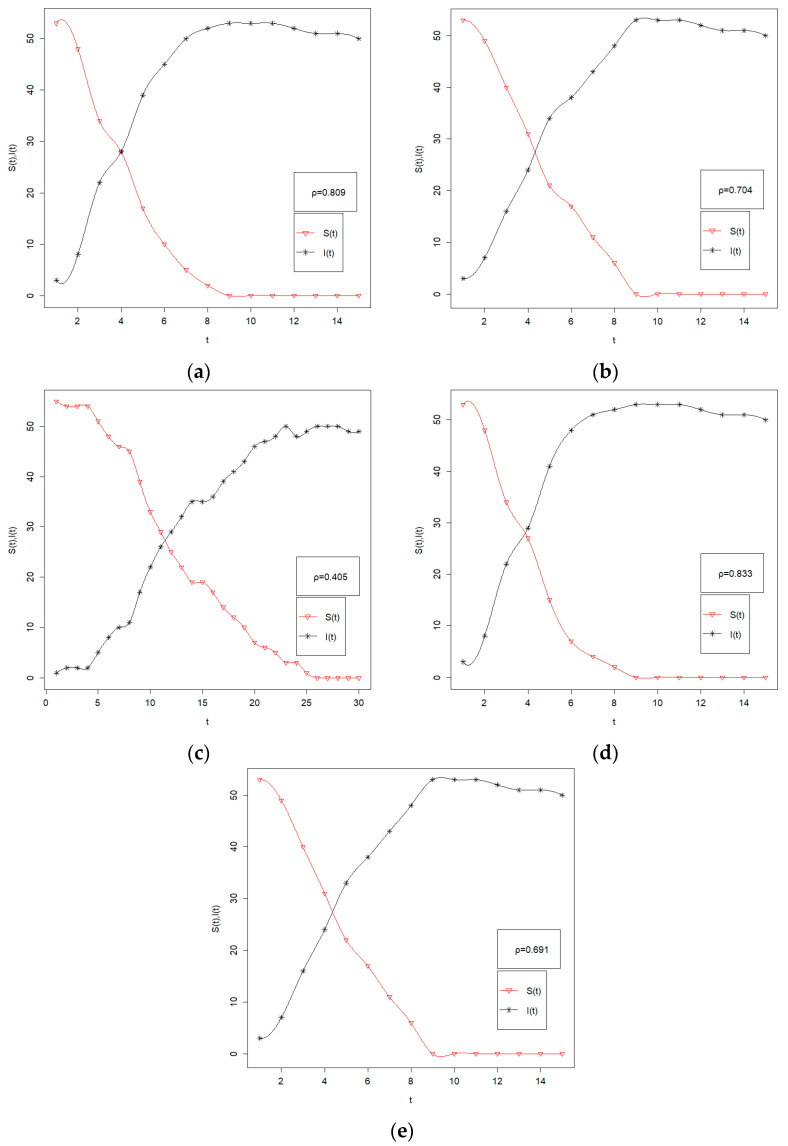
The dynamic evolution of the impact path in five subperiods. (**a**) in stage 1; (**b**) in stage 2; (**c**) in stage 3; (**d**) in stage 4; (**e**) in stage 5

**Table 1 entropy-24-01549-t001:** Descriptive statistics of log returns of 56 listed financial companies.

Company ID	Mean	Std Dev	Min	Max	Skew	Kurtosis	SE	JB	ADF Test
1	−0.00027	0.01925	−0.34079	0.0958	−3.01984	50.745	0.00035	325,158 ***	−15.2 ***
2	−0.00025	0.01723	−0.19705	0.09544	−1.22628	22.034	0.00032	61,207 ***	−14.57 ***
3	0.00030	0.01852	−0.1044	0.09554	0.23946	3.528	0.00034	1580 ***	−13.99 ***
4	−0.00016	0.02371	−0.60744	0.09562	−7.14441	166.674	0.00043	3,484,119 ***	−15.16 ***
5	−0.00020	0.02401	−0.6198	0.09579	−8.01889	193.890	0.00044	4,712,481 ***	−12.78 ***
6	−0.00047	0.01833	−0.21092	0.0958	−2.01532	28.096	0.00034	100,316 ***	−13.58 ***
7	−0.00020	0.01559	−0.10954	0.09625	−0.25476	11.978	0.00029	17,902 ***	−13.88 ***
8	−0.00004	0.01343	−0.1233	0.09531	−0.30273	11.605	0.00025	16,819 ***	−15.12 ***
9	0.00000	0.0154	−0.10577	0.09566	−0.26081	9.377	0.00028	10,986 ***	−13.81 ***
10	−0.00010	0.01352	−0.11629	0.09658	−0.11378	14.146	0.00025	24,926 ***	−13.74 ***
11	−0.00015	0.01906	−0.10564	0.09613	0.34581	6.509	0.00035	5338 ***	−14.15 ***
12	−0.00013	0.02423	−0.54286	0.09563	−4.08067	89.131	0.00044	997,405 ***	−13.34 ***
13	0.00027	0.02271	−0.27236	0.09563	−0.857	13.314	0.00042	22,440 ***	−13.68 ***
14	−0.00033	0.01872	−0.30037	0.0956	−2.67124	42.730	0.00034	230,896 ***	−14.93 ***
15	−0.00061	0.02789	−0.39267	0.09651	−1.84074	27.983	0.00051	99,189 ***	−13.68 ***
16	−0.00015	0.02541	−0.42703	0.0957	−1.45014	29.638	0.00047	110,427 ***	−15.05 ***
17	−0.00031	0.03093	−0.68444	0.09579	−3.61948	81.313	0.00057	829,734 ***	−16.39 ***
18	−0.00049	0.0278	−0.71451	0.0963	−5.62786	147.158	0.00051	2,711,957 ***	−14.92 ***
19	−0.00021	0.02359	−0.10553	0.09576	0.16266	4.334	0.00043	2354 ***	−13.3 ***
20	−0.00023	0.02497	−0.28319	0.09567	−0.3357	10.196	0.00046	13,003 ***	−13.75 ***
21	−0.00023	0.02656	−0.10702	0.09585	0.0826	3.710	0.00049	1719 ***	−14.48 ***
22	−0.00053	0.02932	−0.69188	0.0958	−4.35806	104.728	0.00054	1,375,017 ***	−14.18 ***
23	−0.00035	0.02754	−0.49118	0.0958	−1.97168	36.176	0.0005	164,888 ***	−13.87 ***
24	−0.00038	0.02878	−0.74308	0.09605	−5.69682	149.797	0.00053	2,809,910 ***	−14.62 ***
25	−0.00048	0.03376	−0.10604	0.09646	−0.046	1.806	0.00062	408 ***	−14.34 ***
26	−0.00005	0.02434	−0.79079	0.09545	−11.4086	373.166	0.00045	17,402,008 ***	−14.32 ***
27	−0.00004	0.02199	−0.10544	0.09545	0.07104	2.326	0.0004	677 ***	−14.29 ***
28	−0.00006	0.02223	−0.12358	0.09563	0.39041	3.961	0.00041	2031 ***	−13.61 ***
29	−0.00029	0.02868	−0.71823	0.09679	−5.15102	133.160	0.00052	2,220,858 ***	−14.94 ***
30	−0.0006	0.03112	−0.14812	0.09659	−0.03155	2.031	0.00057	515 ***	−14.24 ***
31	0.00012	0.03071	−0.24064	0.09603	0.18239	3.655	0.00056	1682 ***	−15.32 ***
32	−0.0006	0.02265	−0.10886	0.0992	−0.38755	5.436	0.00041	3756 ***	−15.19 ***
33	−0.00013	0.02964	−0.41689	0.09635	−0.87503	15.464	0.00054	30,163 ***	−15.48 ***
34	0.00005	0.03311	−0.10582	0.09623	0.07885	1.976	0.00061	490 ***	−14.93 ***
35	−0.00029	0.02976	−0.35953	0.096	−1.88577	21.654	0.00054	60,157 ***	−13.94 ***
36	−0.00089	0.03259	−0.54302	0.09671	−1.6518	28.293	0.0006	101,034 ***	−15.03 ***
37	0.00018	0.02754	−0.10558	0.09613	0.25149	2.896	0.0005	1077 ***	−14.24 ***
38	−0.00004	0.03538	−0.6978	0.09604	−2.41286	51.841	0.00065	337,518 ***	−14.99 ***
39	−0.00033	0.03157	−0.65214	0.09572	−3.11113	62.836	0.00058	496,427 ***	−14.03 ***
40	−0.00041	0.02972	−0.56389	0.09659	−2.31462	45.068	0.00054	255,569 ***	−15.09 ***
41	−0.00032	0.03219	−0.10558	0.09599	−0.04499	2.152	0.00059	579 ***	−14.76 ***
42	0.00009	0.03019	−0.10575	0.09635	−0.10173	2.440	0.00055	747 ***	−13.71 ***
43	−0.0002	0.0252	−0.25874	0.0958	−0.38966	6.841	0.00046	5906 ***	−13.96 ***
44	−0.00021	0.03509	−0.72762	0.09566	−3.10695	62.533	0.00064	491,681 ***	−12.95 ***
45	−0.00023	0.02832	−0.10572	0.09659	−0.00914	3.434	0.00052	1470 ***	−14.68 ***
46	−0.00064	0.02934	−0.72177	0.09764	−4.88404	123.482	0.00054	1,910,299 ***	−14.24 ***
47	−0.00028	0.02812	−0.10603	0.09675	0.04238	3.078	0.00051	1182 ***	−14.67 ***
48	−0.00047	0.03314	−0.70163	0.09685	−6.1338	128.233	0.00061	2,066,026 ***	−13.73 ***
49	0.0002	0.03236	−0.38446	0.0959	−0.54472	8.284	0.00059	8695 ***	−14.61 ***
50	−0.00038	0.02802	−0.35474	0.0959	−0.80014	11.282	0.00051	16,171 ***	−13.91 ***
51	−0.00012	0.02716	−0.10568	0.09563	−0.26153	3.416	0.0005	1488 ***	−14.74 ***
52	−0.00012	0.03092	−0.69084	0.09566	−3.75775	84.529	0.00057	896,650 ***	−14.06 ***
53	−0.00006	0.03129	−0.60268	0.09612	−2.32364	47.531	0.00057	283,982 ***	−14.75 ***
54	−0.00042	0.02932	−0.4416	0.09583	−1.51514	24.655	0.00054	76,834 ***	−15.19 ***
55	0.00007	0.04142	−0.75922	0.16436	−4.69051	83.319	0.00076	875,291 ***	−13.79 ***
56	−0.00052	0.03658	−0.89126	0.09638	−8.88564	208.805	0.00067	5,467,534 ***	−14.95 ***

Note: *** *p* < 0.01.

**Table 2 entropy-24-01549-t002:** Time series observations in each substage.

Sub-Stage	Starting Time	Ending Time	Typical Extreme Events and Stage Characteristics
Stage 1	4 January 2010	21 October 2013	Post-financial crisis and 2010 European debt crisis
Stage 2	22 October 2013	13 January 2017	2013 money crisis and 2015 stock market crash in China
Stage 3	14 January 2017	10 April 2018	Stabilization period
Stage 4	11 April 2018	23 October 2020	US-China trade conflict and COVID-19 epidemic
Stage 5	24 October 2020	18 April 2022	Post-COVID-19 era

**Table 3 entropy-24-01549-t003:** Spatial correlation test.

Spatial Effect Test	*W_gene_*	*W_area_*	*W_p_*
Moran’s I	0.237 ***	0.239 ***	0.356 ***
Z(I)	65.717	65.54	161.58
Geary C	0.481 ***	0.478 ***	0.656 ***
LMlag	8970.12 ***	8732.40 ***	19665.95 ***
LMerro	9128.71 ***	8868.87 ***	22966.41 ***
robust LMlag	159.19 ***	157.45 ***	66.38 ***
robust LMerro	318.04 ***	293.92 ***	3366.84 ***

Note: *** *p* < 0.01.

**Table 4 entropy-24-01549-t004:** Variables list.

Variable Name	Symbol	Definition
Log return rate	*r*	Daily log return rate of listed financial institutions
Turnover rate	*turnover*	Turnover rate of circulating capital stock of financial institutions
Low-volatility dummy variable	*lvar*	Takes 0 for the high-volatility group and 1 for the low-volatility group, and the value for the middle groups remains unchanged
High-volatility dummy variable	*hvar*	Takes 1 for the high-volatility group and 0 for the low-volatility group, and the value for the middle groups remains unchanged
Exchange rate	*Erate*	Change rate of daily central parity rate of RMB against USD
Interest rate	*DR001*	Change rate of weighted average interest rate of overnight repo between banks with interest rate bonds as collateral

**Table 5 entropy-24-01549-t005:** Regression results of the spatial lag model.

	Stage 1			Stage 2			Stage 3			Stage 4			Stage 5		
	*W_gene_*	*W_area_*	*W_P_*	*W_gene_*	*W_area_*	*W_P_*	*W_P_*	*W_area_*	*W_P_*	*W_gene_*	*W_area_*	*W_P_*	*W_gene_*	*W_area_*	*W_P_*
*λ*	0.808 *** (328.29)	0.809 *** (344.56)	0.806 *** (177.54)	0.703 *** (154.99)	0.704 *** (161.13)	0.796 *** (154.95)	0.383 *** (25.27)	0.405 *** (28.37)	0.661 *** (47.86)	0.829 *** (282.38)	0.833 *** (296.05)	0.833 *** (174.59)	0.680 *** (92.10)	0.691 *** (98.11)	0.756 *** (83.34)
*turnover*	0.0023 *** (35.43)	0.0022 *** (35.44)	0.0024 ** (37.28)	0.0019 *** (29.15)	0.0019 *** (29.12)	0.0019 *** (29.95)	0.0012 *** (11.37)	0.0012 ** (11.43)	0.0012 ** (12.51)	0.0017 *** (29.66)	−0.0017 *** (29.59)	0.0015 *** (27.56)	0.0016 *** (15.63)	0.0017 *** (15.59)	0.0016 *** (15.36)
*lvar*	0 (−0.25)	0.00007 (−0.26)	0.00015 (0.57)	−0.00045 (1.23)	−0.00043 (1.20)	−0.00045 (1.29)	0.00036 (0.74)	0.0004 (0.75)	0.00026 (0.55)	−0.00063 ** (−1.78)	−0.00061 ** (−1.72)	−0.00034 (−0.98)	−0.00032 (0.67)	−0.00033 (0.68)	0.00021 (0.45)
*hvar*	−0.0025 *** (−9.23)	−0.0025 *** (−9.24)	−0.0026 *** (−10.30)	−0.0020 *** (−5.37)	−0.0020 *** (−5.37)	−0.0020 *** (−5.60)	−0.0016 *** (−3.35)	−0.0017 *** (−3.35)	−0.0018 *** (−3.81)	−0.0022 *** (−6.68)	−0.0022 *** (−6.61)	−0.0022 *** (−7.14)	−0.0027 *** (−6.54)	−0.0028 *** (−6.54)	−0.0027 *** (−6.64)
*Erate*	−0.406 (−0.67)	−0.394 (−0.65)	−1.817 *** (−3.21)	−0.254 (0.98)	2.604 (1.00)	−0.099 (−0.28)	−0.089 (−0.65)	0.090 (−0.63)	−0.030 (−0.13)	−0.802 ** (−2.52)	−0.815 ** (−2.51)	−0.43 * (−1.46)	−0.346 * (1.69)	−0.361 ** (1.70)	−0.231 (−0.89)
*DR001*	−0.0104 ** (−2.38)	−0.0104 *** (−2.36)	−0.0047 (−1.14)	−0.0379 *** (−3.36)	−0.038 *** (−3.36)	−0.022 * (−1.45)	−0.009 ** (−1.74)	−0.009 *** (−1.71)	−0.007 (−0.79)	0.015 *** (3.59)	0.019 *** (3.62)	0.0015 (−0.33)	0.0003 (0.11)	0.0003 (0.13)	−0.0039 (−1.11)
*φ*	5.208333	5.235602	5.154639	3.367003	3.378378	4.901961	1.620746	1.680672	2.949853	5.847953	5.988024	5.988024	3.125	3.236246	4.098361
Institution effect	Yes	Yes	Yes	Yes	Yes	Yes	Yes	Yes	Yes	Yes	Yes	Yes	Yes	Yes	Yes
Observations	51,240	51,240	51,240	44,408	44,408	44,408	16,744	16,744	16,744	34,552	34,552	34,552	20,160	20,160	20,160

Note: (1) Spatial effect multiplier *φ* = 1/(1 − *λ*); (2) * *p* < 0.1; ** *p* < 0.05; *** *p* < 0.01.

**Table 6 entropy-24-01549-t006:** Bonacich key nodes.

Rank	Stage 1	Stage 2	Stage 3	Stage 4	Stage 5
1	BJT (6.625)	DFH (1.240)	DXC (1.504)	DAX (1.065)	ST.DLT (2.564)
2	BMS (6.463)	DHD (0.872)	ST.DLT (1.115)	ST.DLB (0.901)	ST.DLB (2.352)
3	BGS (6.270)	DHH (0.812)	DSG (1.093)	ST.SAX (0.813)	DPF (1.823)
4	BJS (6.145)	BPF (0.643)	ST.DLB (1.077)	DJW (0.730)	IXS (1.752)
5	BBJ (5.682)	DAX (0.620)	ST.DPF (1.002)	ST.DLT (0.582)	DOW (1.664)
6	BPF (5.672)	BGS (0.597)	DOW (0.983)	DHD (0.468)	DHD (1.642)
7	BHX (5.400)	DPB (0.515)	DJW (0.953)	DAJ (0.349)	IHB (1.629)
8	DYX (5.154)	BIB (0.499)	DKW (0.825)	ST.DPF (0.310)	DKW (1.151)
9	BIB (5.071)	BBJ (0.475)	DMI (0.810)	ST.DWR (0.247)	BMC (1.146)
10	DLV (4.855)	DMI (0.430)	BCH (0.755)	DFH (0.215)	DLV (1.089)

Note: Symbols beginning with the letters ST indicate that the listed institution faces the possibility of delisting.

## Data Availability

Data will be made available on request.

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
