# Peer review of "Spatial Spillovers of Financial Risk and Their Dynamic Evolution: Evidence from Listed Financial Institutions in China"

_entropy, 2022, doi:10.3390/e24111549_

Round 1

Reviewer 1 Report

The authors introduced an interesting topic by using the current available methods. But this journal is based on new methodology paper, I think. The authors need to propose new method for your research topic. I do not find any significant methods from your papers. It is just a mixed presentation of well known statistics methods. I suggest the authors to submit your paper to applied science, https://www.mdpi.com/journal/applsci, or sustainability, https://www.mdpi.com/journal/sustainability .

Author Response

We thank the reviewer for the comments. Our paper not only applies statistical methods in the study but also but also enriches and extends the existing literature. The paper extends the spatial spillover effects of financial risks existing in the the regional economies to the field of financial institutions. Furthermore, the paper applies the complex network theory to the field of spatial econometrics. While most existing studies are limited to a single spatial effect, we construct the economic measure distance and gravitational spatial weight matrix by considering the geographical distance and financial correlation coefficients to explore its multidimensional spatial spillover effects and spillover paths. We believe our paper presents a new perspective on the spatial financial risks by using an innovative approach.

Reviewer 2 Report

This paper uses a range of econometric methods to study spillovers of financial risk between listed financial institutions in China. The results show that the multidimensional spatial spillovers of financial risk among financial institutions are evident and time-varying. The spatial spillovers of financial institutions are positively correlated with the turnover rate and negatively correlated with the exchange rate, interest rate, and return volatility.

 In order for the article to be more complete, the following shortcomings need to be clarified:

1.     In terms of studying samples, the explanation for sample selection is necessary for a proper implication.

1.1. China has 6 state-owned commercial banks and 12 national commercial banks, however, there are only 14 of them being used in the studying samples.

1.2. According to the China Securities Regulatory Commission in 2021, there are 103 securities companies.

1.3. There are 14 insurance companies in China, including Anbang, Sinosure, China Pacific Insurance, Company, China Re, China Taiping Insurance Holdings, China United SME, Guarantee Corporation, Guohua Life, Ming An Holdings, New China Life Insurance, People's Insurance Company of China, PICC Property and Casualty, Ping An Insurance, Tianan Insurance, ZhongAn.

1.4. etc.

2.     Table 1 is recommended to move to Appendix.

3.     Line 326: “Note: * p < 0.1; ** p < 0.05; *** p < 0. 1” and Table 6: “Note: (1) Spatial effect multiplier φ=1/(1-λ); (2) * p < 0.1; ** p < 0.05; *** p < 0.1.” , *** p < 0. 1 should be revised as *** p < 0.01

4.     Is the stationary test necessary for all variables? If not, it is recommended that the authors should explain why it is not essential.

5.     Line 334-338: The correlation between Fin300 and HS300 is used to measure the risk spillover from China's financial sector market to the overall economy, which is fitted by an ARMA (2,2)-GARCH (1,1). It is recommended that the authors should explain why ARMA(2,2)-GARCH(1,1) is superior to other models, like ARMA(2,1)-GARCH(2,1) or TGARCH (1, 1), etc.

6.     There are 4 breakpoints detected in the dynamic conditional correlation coefficients from January 5, 341 2010 to April 18, 2022. Are these breakpoints consistent with those of individual stock returns? If not, what are the explanations for using these correlation coefficients in the multidimensional economic spatial regression model?

7.     Multidimensional economic spatial regression model: Line 229-231 states “the general stock weighted return of the financial market (λgeneWgene), the stock weighted return of across regions (λareaWarea) and the stock weighted return rate across political administrative regions (λpWp).”, I wonder how the economic distance measure can affect the daily returns of financial institutions?

8.     Authors should add an explanation of what is the advantage of spatial spillovers of financial risk over other conventional spillover effect methods. Why use a more complex approach to solve a problem when there is a more straightforward solution?

Author Response

  1. In terms of studying samples, the explanation for sample selection is necessary for a proper implication.

1.1. China has 6 state-owned commercial banks and 12 national commercial banks, however, there are only 14 of them being used in the studying samples.

1.2. According to the China Securities Regulatory Commission in 2021, there are 103 securities companies.

1.3. There are 14 insurance companies in China, including Anbang, Sinosure, China Pacific Insurance, Company, China Re, China Taiping Insurance Holdings, China United SME, Guarantee Corporation, Guohua Life, Ming An Holdings, New China Life Insurance, People's Insurance Company of China, PICC Property and Casualty, Ping An Insurance, Tianan Insurance, ZhongAn.

1.4. etc.

Thanks for the reviewer’s comment. Since China’s public listing financial market started fairly recently, most financial institutions were listed only within a few years. According to the information disclosed by the China Securities Regulatory Commission, as of the third quarter of 2021, there were 126 public listed financial institutions in China, including 41 banks, 34 securities institutions, 7 insurance companies, and 44 diversified financial institutions. Considering the requirement of the size and the timeframe of the data set, we select 56 financial institutions listed before January 2010 as the sample, including 14 banks , 11 securities companies, 5 insurance companies, and 26 diversified financial institutions, which is a highly representative sample. See the first paragraph of Section 3.1.

  1. Table 1 is recommended to move to Appendix.

Table 1 has been placed in Table E1 in the Appendix.

  1. Line 326: “Note: * p < 0.1; ** p < 0.05; *** p < 0. 1” and Table 6: “Note: (1) Spatial effect multiplier φ=1/(1-λ); (2) * p < 0.1; ** p < 0.05; *** p < 0.1.” , *** p < 0. 1 should be revised as *** p < 0.01

We thank reviewer for careful review. The errors were corrected in accordance with reviewer’s comment. See the Table 1 of Section 3.1 and Table 5 of Section 3.4.

  1. Is the stationary test necessary for all variables? If not, it is recommended that the authors should explain why it is not essential.

We thank the reviewer for the insightful comment. We have added the results of the ADF stationary test to the descriptive statistics results and all variables are stationary at the 1% significance level. See the second paragraph of Section 3.1 of the revised manuscript.

  1. Line 334-338: The correlation between Fin300 and HS300 is used to measure the risk spillover from China's financial sector market to the overall economy, which is fitted by an ARMA (2,2)-GARCH (1,1). It is recommended that the authors should explain why ARMA(2,2)-GARCH(1,1) is superior to other models, like ARMA(2,1)-GARCH(2,1) or TGARCH (1, 1), etc.

With reference to the reviewer’s comment, we have added a rationale regarding the choice of the ARMA (2,2)-GARCH (1,1) model. Studies have shown that GARCH models can effectively characterize the returns of financial assets, and the simplest GARCH (1,1) model can capture the characteristics of financial asset returns. According to the AIC criterion, the optimal lag order of the ARMA model is chosen as (2,2). See the first paragraph of Section 3.2 of the revised manuscript.

  1. There are 4 breakpoints detected in the dynamic conditional correlation coefficients from January 5, 341 2010 to April 18, 2022. Are these breakpoints consistent with those of individual stock returns? If not, what are the explanations for using these correlation coefficients in the multidimensional economic spatial regression model?

We thank the reviewer for the insightful comment. In this paper, we use t-copula-DCC-GARCH to measure the dynamic correlation between the Chinese financial submarkets and the national stock market and then perform the Bai & Perron structural breakpoint test. This is to identify abnormal fluctuations in the financial market through objective data and then to stage them scientifically. It should be noted that the dynamic correlation coefficients are not used in the multidimensional economic spatial regression model. Instead, the static tail correlation coefficients Ri,j for different periods are measured using the t-copula-GARCH model and then the generalized economic distance is constructed by the static correlation coefficients. Finally, the spatial spillover effects of financial risks are analyzed. The lack of clarity in the original text caused misunderstanding to the reviewers. We have added it in the second paragraph of Section 2.2.1 of the revised manuscript. “Ri,j is the static correlation coefficient measured by the t-copula-GARCH model, representing the tail correlation between financial institutions i and j”.

  1. Multidimensional economic spatial regression model: Line 229-231 states “the general stock weighted return of the financial market (λgeneWgene), the stock weighted return of across regions (λareaWarea) and the stock weighted return rate across political administrative regions (λpWp).”, I wonder how the economic distance measure can affect the daily returns of financial institutions?

We added the way in which EDM affects the daily returns of financial institutions. According to the definition of the EDM and the gravitational effect spatial weights matrix W, it can be concluded that the EDM is negatively correlated with wi,j and positively correlated with mimj. From the perspective of Wgene, mi and mj (market capitalization of financial institutions) reflect the prior market performance of financial institutions. Since the innate profit-seeking nature, financial institutions always chase financial assets with good market performance and therefore, the returns of financial institutions with good prior market performance usually exhibit the same volatility trend. From the perspective of Wp, the EDM between financial institutions in the same administrative region is smaller than that between financial institutions in different administrative regions. Financial institutions in the same administrative region are susceptible to the same regional policies, which may lead to similar volatility in their daily returns. With regard to Warea, the greater the correlation Ri,j between two financial institutions, the closer their business crossover and business transactions are likely to be, and thus, their daily returns will generate the same volatility. See the fourth paragraph of Section 2.3 of the revised manuscript.

  1. Authors should add an explanation of what is the advantage of spatial spillovers of financial risk over other conventional spillover effect methods. Why use a more complex approach to solve a problem when there is a more straightforward solution?

We thank the reviewer for the comment. Conventional spillover effect methods mainly measure spillover effects based on market information among financial assets. These methods ignore the influence of spatial factors including distance and region on financial risk and do not consider the multidimensional spillover path of financial risk, which lacks the accuracy of identifying spatial spillover of financial risk. It is necessary and important to use new methods to measure the spatial spillover of financial risk, which can capture multidimensional spatial spillover characteristics. See the sixth paragraph of Section 1 of the revised manuscript.

Reviewer 3 Report

The contribution of this work is very good. The title has been formulated
unambiguously conveying the focus of the study.
The accurate interpretation of outcomes, well substantiated by the results of
the analysis h as been achieved by them. The presentation of the results in
terms of the research objectives has been successfully made.

The authors have been able to draw logical conclusions from the results. The simulations results and their respective discussion points prove the efficacy of the proposed method. Conclusions are accurate and clearly based on outcomes. Appropriate research goals are chosen in this contribution, which shows that the a u thors have a high level of understanding of
current research within the field of their research. The authors successfully used the appropriate techniques for analysis of the research objects.

The unique thing to be clarified is the concept of "The tail risk network".

Please write to clarify the concept better.

Concerning the cited literature you can consider the following paper s to improve the tutorial aspects of the paper.

Chen, H. et al. Extension of SEIR Compartmental Models for Constructive Lyapunov Control of COVID-19 and Analysis in Terms of Practical Stability. Mathematics 2021, 9, 2076. https://doi.org/10.3390/math9172076

Simulation of SARS-CoV-2 pandemic in Germany with ordinary differential equations in MATLAB

Jakob Wieland et al.

2021 25th International Conference on System Theory, Control and Computing (ICSTCC)   Wu, F.; Zhang, Z.; Zhang, D.; et al. Identifying systemically important financial institutions in China: New
evidence from a dynamic copula-CoVaR approach. Annals of Operations Research. 2021, 1-35.

Author Response

The unique thing to be clarified is the concept of "The tail risk network". Please write to clarify the concept better.

Following the reviewer’s comment, we further elaborate the tail risk network. We construct the tail risk network with financial institutions as nodes and the tail correlation between financial institutions as edges. The institutional nodes in the tail risk network play the role of both the receiver and the transmitter of risk, which effectively describes the contagious financial risk among financial institutions. See the first paragraph of Section 2.4.1 of the revised manuscript.

Concerning the cited literature you can consider the following paper s to improve the tutorial aspects of the paper.

We have added to the recommended literature. Wu et al. (2021) has been presented in the original manuscript and the other two papers are added in the Introduction section. See the seventh paragraph of Section 1 of the revised manuscript.

Round 2

Reviewer 1 Report

The paper needs to add more case studies to the current paper to verify the proposed method.

Author Response

Dear Reviewer,

We have added more discussions related to the results in Section 3.4. Please see the highlighted addition. Thank you!

Reviewer 2 Report

Dear Authors,

I carefully reviewed the responses given. I think the authors have fully addressed my concerns, and recommend an acceptance.

Author Response

We sincerely thanks the reviewer for the careful review and insightful comments.